# Scanning Electron Microscopy and EDX Spectroscopy of Commercial Swabs Used for COVID-19 Lateral Flow Testing

**DOI:** 10.3390/toxics11100805

**Published:** 2023-09-24

**Authors:** Manuel Aparicio-Alonso, Verónica Torres-Solórzano, José Francisco Méndez-Contreras, Karina Acevedo-Whitehouse

**Affiliations:** 1Medical Direction and Healthcare Responsibility, Centro Médico Jurica, Santiago de Querétaro 76100, Mexico; 2Unit for Basic and Applied Microbiology, Universidad Autónoma de Querétaro, Santiago de Querétaro 76140, Mexico; vtorres23@alumnos.uaq.mx; 3Petrochemical Complex Production, Petróleos Mexicanos, Veracruz 96400, Mexico; jfmendez96@hotmail.com

**Keywords:** nasopharyngeal swabs, COVID, energy dispersive X-ray spectroscopy, scanning electron microscopy, transition metals, metalloids, titanium, zirconium, aluminium, silicon

## Abstract

The chemical composition of COVID test swabs has not been examined beyond the manufacturer’s datasheets. The unprecedented demand for swabs to conduct rapid lateral flow tests and nucleic acid amplification tests led to mass production, including 3D printing platforms. Manufacturing impurities could be present in the swabs and, if so, could pose a risk to human health. We used scanning electron microscopy and energy dispersive X-ray (EDX) spectroscopy to examine the ultrastructure of seven assorted brands of COVID test swabs and to identify and quantify their chemical elements. We detected eight unexpected elements, including transition metals, such as titanium and zirconium, the metalloid silicon, as well as post-transition metals aluminium and gallium, and the non-metal elements sulphur and fluorine. Some of the elements were detected as trace amounts, but for others, the amount was close to reported toxicological thresholds for inhalation routes. Experimental studies have shown that the detrimental effects of unexpected chemical elements include moderate to severe inflammatory states in the exposed epithelium as well as proliferative changes. Given the massive testing still being used in the context of the COVID pandemic, we urge caution in continuing to recommend repeated and frequent testing, particularly of healthy, non-symptomatic, individuals.

## 1. Introduction

Methods to diagnose infectious diseases increasingly depend on rapid tests that allow the detection of specific antigens or antibodies against a specific antigen. It is beyond the scope of this paper to discuss the validity of equating the detection of a protein fragment to diagnosing disease. Regardless of the nuances in such assumptions, these rapid tests are currently considered central for disease diagnosis. In the context of the COVID-19 (herein, COVID) pandemic, nucleic acid amplification tests (NAAT) and lateral flow tests (LFT), both based on nasopharyngeal swabbing, are the official methods for diagnosing COVID, and since their authorization at the start of 2020, thousands of millions of tests have been performed.

According to Our World in Data (accessed on 15 December 2022), between 1 January 2020 and 16 December 2022, 5,073,521,930 COVID tests were performed, and this amount does not include the home tests that can be purchased in many countries and that are not informed to the health authorities, particularly when negative. The demand for such large-scale testing for COVID, unsurprisingly, could not be met by ordinary manufacturing of commercial nasopharyngeal swabs [1]. This led to mass production of swabs across various established and emerging manufacturers, and even to the three-dimensional (3D) printing of nasopharyngeal swabs [2,3]. It is plausible that owing to the unprecedented speed of manufacturing and the novel technological approaches to do so, impurities could realistically be present in the swabs used for mass testing. In the European Union, production, and commercialization of swabs for diagnostic purposes requires a *Conformité Européenne* (CE), certificate, which confirms that the manufacturer meets the minimum criteria for safety and environmental protection, in accordance with the provisions of Directive 93/42/EEC (14 June 1993) and Regulation 2017/745 (5 April 2017). However, to the best of our knowledge, the technical datasheets for the COVID nasopharyngeal swabs do not provide any information on the presence or maximum permitted amounts of microscopic and nanoscopic elements.

In terms of their fabric, there are different types of swabs used for diagnostic purposes, including cotton, rayon, polyester, nylon, polyurethane and foam [4,5,6]. Physical and chemical properties of the fabric can influence sample adhesion and release [7], owing to structural variations of the fabric [8]. Interestingly, an unbiased search of the literature reveals that not a single study on the composition of the COVID test swabs has been published. Nearly all the published studies on nasopharyngeal swabs used for COVID testing have examined their performance [4,7,9,10] and we found only one review paper that compared the physical properties of different swab fabrics [11].

In order to obtain confident information about the safety of the COVID swabs, it is crucial to know their components, particularly as the testing method demands insertion of the swab into the delicate nasopharyngeal anatomy. Furthermore, despite nasopharyngeal swabs being graded as class I medical devices [12], and as such, being considered low risk, swabbing complications are known to occur, albeit at a low frequency [13]. There are now published reports of complications stemming from COVID testing, that range from mild (i.e., discomfort, pain or bleeding), to moderate (i.e., ethmoidal silent sinus syndrome), and serious, including breaches of the anterior skull base associated with a risk of meningitis [12,14,15,16,17,18]. If impurities are present in the swab, the risk of exposing the nasopharyngeal epithelium, or, even, the bloodstream to these should not be ignored, particularly considering repeated exposure from recurrent sampling.

This study is the first report on the ultrastructure and chemical composition of seven brands of COVID testing swabs that were selected at random among commercially available tests in the country where our study took place.

## 2. Materials and Methods

We analysed two to three specimens of each of seven brands of nasopharyngeal swabs: iHealth, Puritan HydraFlock, MANTACC, Nasal Swab, FLOQSwabs, Kangdaan, and Taizhou (Table 1). These brands were selected at random from the COVID test kits that were available for purchase. In addition, one specimen of two cotton swab brands commonly used as applicators and bacterial transport was included as control swabs (not COVID-test swabs). To determine swab morphology, scanning electron microscopy (SEM) images of the tip, lateral walls and base of the swabs were taken using an EVO50 (Carl Zeiss AG, Jena, Germany) SEM with variable pressure and an acceleration voltage of 20 kV and with magnification at 30× and 100×. Resolution was 200 µm and 100 µm. To identify the chemical composition of the swabs’ fabric, we performed X-ray spectroscopy using SE1, BSD, and EDX detectors. All analyses were conducted at the Laboratory of Microscopy of the School of Natural Sciences of the Autonomous University of Queretaro (Mexico). All swabs were handled with sterile gloves, and the packages were opened with extreme care immediately before the SEM and EDX spectroscopy to avoid potential contamination of the swabs.

We weighed the head of each of the swab brands to determine their mass (to the nearest 0.001 g). These data were used to convert the percentage of each element detected by EDX spectroscopy to mg, by way of a direct mathematical proportion, considering the sum of the percentage of each element detected (range: 99.02% to 99–79%) as the total percentage.

## 3. Results

SEM images of the swabs are shown in Figure 1. Two of each of the different swab brands made of foam (1), nylon (1), flocked nylon (5) and one of each control (cotton swabs not used for COVID testing) swabs were examined by SEM. The integrity of the head appeared intact, without evidence of biological (fungal spores or bacteria), contamination. However, some of the flocked nylon swab brands showed crust-like elements surrounding the fibres. The iHealth foam swabs (Figure 1A) revealed an alveolus-like network structure of varying diameters. In the flocked nylon swabs (Figure 1B–E), individual fibres of homogeneous length were observed protruding from the head of the swab, more noticeably so in the HydraFlock swab, where each fibre had split or unravelled ends (Figure 1B), similar to the head of a Hydra, which is where the swab brand derives its name. In the cotton swabs (Figure 1F,G), fibres were disorganized and with varying lengths, wrapped in one single direction.

Eleven elements were detected in the EDX spectroscopy analysis of the swabs (Table 2 and Appendix A), of which the most abundant were carbon (comprising 48.97% to 63.84%) and oxygen (comprising 23.24% to 50.31%). Nitrogen was detected in all but one of the seven COVID test swab brands (ranging from 4.75% to 10.26%), and we detected fluorine (F), silicon (Si), titanium (Ti), strontium (Sr), aluminium (Al), zirconium (Zr), gallium (Ga), and sulphur (S) at low percentages (0.03–1.2%). Of these, Si, Zr, Sr, Ga, Al, and S were not detected in the sterile cotton swabs used for generic applications, such as applicators and for transport of bacterial samples, and not for COVID testing. The amounts of the elements identified were variable and, in some cases, were detected in only one or two of the replicates examined for each swab brand.

## 4. Discussion

Having knowledge about the chemical composition of the fabric and potential manufacturing impurities or by-products present in nasopharyngeal swabs that are being so frequently used for COVID testing of humans and, even, animals is essential to ensure that potential health problems are minimized [19,20,21]. We conducted a pilot study with a limited number of samples to examine the chemical composition and ultrastructure of seven brands of common COVID testing swabs. The methods used are robust, and while it could be argued that other techniques, such as Raman spectroscopy or Fourier Transform Infrared Spectroscopy (FTIS), would allow us to map the distribution of the chemical elements detected in the nasopharyngeal swabs, these approaches fell outside the scope of our study. Future studies could aim to include such analytical techniques to expand on our findings.

Hydrogen atoms are not detectable by X-ray spectroscopy [22], but most of the elements detected, carbon, oxygen and nitrogen, concur with what is expected to be found when analysing the composition of biopolymers and synthetic polymers. Specifically, cotton, rayon and polyester polymers are formed by carbon, hydrogen, and oxygen, with cotton and rayon having the formula (C_6_H_10_O_5_)_n_ [23], and polyester (C_10_H_8_O_4_) [24]. Nylon polymers in turn, are formed by carbon, hydrogen, oxygen and nitrogen, (C_12_H_22_N_2_O_2_)_n_ [25] and foam swabs are made of high-density polyurethane elastomers formed by diols (HO-R-OH) and diisocyanate (NCO-R′-NCO) [26]. However, in addition to those expected elements, we found evidence of metalloids (silicon), transition metals (titanium and zirconium), post-transition metals (aluminium and gallium), and alkaline earth metals (strontium) in some of the swabs, although none of these elements are indicated in the technical datasheet of the manufacturers. We also found evidence of traces of fluorine in five of the examined brands, and traces of gallium and sulphur were present in one of the brands examined.

Evidently, it is the dose that makes the poison, and the unexpected elements detected by EDX spectroscopy were found in relatively small quantities. However, their presence certainly warrants a discussion in terms of their potential impact human health, particularly given the high number of COVID tests that many people are undergoing, which could potentially lead to bioaccumulation from repeated exposure. The bioaccumulation of aluminium [27,28] and fluoride are well-recognized phenomena [29], and are highly dose-dependent [30,31]. Less studied is the bioaccumulation of transition metals in humans. However, experimental studies have shown that titanium nanoparticles can accumulate in tissues following contact with epithelia [32,33], even at low concentrations [34], and zirconium oxide (ZrO_2_), commonly used in biomedical and dentistry applications [35,36], has shown evidence of bioaccumulation in aquatic animals after environmental exposure at low concentrations [37]. Silicon dioxide nanoparticles (n-SiO_2_), among the most commonly used nanomaterials in biomedicine, pharmaceutical manufacturing [38,39,40], and cosmetics, are increasingly being released into the environment, and there is experimental evidence that they accumulate in the food chain [41,42]. Under the precautionary principle of medicine, it would be wise to assume that these elements, detected in the COVID test swabs, could be accumulated in human tissues.

The biological impact of the detected elements also warrants discussion. We will start by addressing the presence of fluorine. According to the Agency for Toxic Substances and Disease Registry (ATSDR) of the Health and Human Services Department of the US, fluorine is soluble fluoride and is a naturally occurring, widely distributed element. The elemental form of fluorine is extremely reactive, and when in contact with water, forms fluorides and hydrofluoric acid [43]. Various studies have shown that fluoride can induce oxidative stress, deregulate cellular redox potential, lead to mitochondrial damage, promote endoplasmic reticulum stress and alter gene expression [44]. Studies on mice have shown that pre and perinatal exposure to fluoride can lead to neurobehavioral alterations, oxidative stress in the brain, and alteration of cholinergic and glutamatergic enzymes [45]. According to the US Environmental Protection Agency, the no observed adverse effect level (NOAEL) of fluoride is 1 ppm (converted to 0.06 mg/kg/day; accessed on 15 December 2022 https://iris.epa.gov/static/pdfs/0053_summary.pdf). Given the amount of fluorine detected in the COVID test swabs (average: 3.316 mg/g, representing 0.17 mg in the swab head), if two swabs were used consecutively on a new-born baby that weighed less than 4 kg, fluorine exposure would exceed the daily NOAEL. Evidently, we have no way of knowing how much of the fluoride in the swabs would actually be transferred to the nasopharyngeal epithelium, but given the amount detected, it would be relevant for future studies to examine trace amounts of fluoride in the nasopharyngeal epithelium after swabbing.

Aluminium was detected in three of the COVID test swabs. Analysis of more samples would be necessary to establish whether its presence is due to the manufacturing process or is an unintended contaminant during swab packaging. It could be argued that if swabs contain aluminium, the amount present is lower than the LOAEL of 26 mg/kg/day (oral route of administration) [46], making it unlikely that there would be any harmful effect from exposure. However, nasal exposure to aluminium salts (in rats) has shown that even small doses (10 µg) can lead to limited damage in the olfactory epithelium and detectable levels of aluminium in the olfactory bulb [47]. Neurotoxicity of aluminium via inhalation has been studied and tends to be associated with higher doses; however, nasal instillation of rats with even low amounts (as low as 1 mg/kg) of Al_2_O_3_, revealed dose-dependent inflammation and alveolar–capillary barrier permeabilization after exposure [48]. Furthermore, aluminium can penetrate cell membranes in the site of exposure and travel via the bloodstream to enter other cells, where it binds to proteins and enzymes, modulating cytokine expression [49,50]. If this were to occur following exposure to aluminium in the swabs, susceptible people could plausibly experience overexpression of cytokines in the nasopharyngeal mucosa [51], as has been described for aluminium adjuvants in nasal vaccines [52].

Silicon was another unexpected finding, with amounts varying from 5.78 to 49.66 mg/g. Data on the lowest estimated cumulative exposure range that has been reported in the literature are established for silica, a common oxide form of silicon, at <0.2 mg/m^3^/year [53], the equivalent of to up to 0.2 mg/kg/year. If we consider the highest value of silicon detected in one of the COVID test swab brands analysed here (Taizhou^®^, 49.66 mg/kg, representing 2.55 mg in the swab head), undergoing seven to eight COVID test swabbing events in one year (one test every 48 days) would exceed the lowest estimated cumulative exposure range of silicon for an adult of 70 kg, and the impact would be even worse for children or babies, which have been sampled even as early as 5 days from birth [54]. The toxic effects of cumulative inhalation of silica are not trivial. Death due to silicosis was observed in 3.8% of mining and pottery workers in China, all of which were cumulatively exposed to a range of 0.1–1.23 mg/m^3^/year [55]. Nanoparticles of silicon dioxide between 30 nm and 1000 nm in diameter can induce extremely high levels of expression of the pro-inflammatory cytokine IL-1β, cause lysosomal instability, increase reactive oxygen species (ROS) levels, and lead to cell death in mouse models [56]. Actually, it has been claimed that silica nanoparticles, commonly found in a number of diagnostic and therapeutic applications, when inhaled show mostly reversible pulmonary inflammation, granuloma formation and localized emphysema [57]. Finding silica in three of the analysed swab brands justifies further studies to expand our understanding of the potential biological effects of silica nanoparticles in the nasal epithelium [57].

It is not clear what the potential toxicity of strontium would be for nasal exposure. Most of the toxicological profiles have been determined for radioactive strontium (^90^Sr), used for medical diagnostic procedures [58,59], and for strontium compounds, such as strontium peroxide (SrO_2_), strontium arsenite (As_2_O_4_Sr), strontium nitrate (Sr(NO_3_)_2_) and strontium chromate (SrCrO_4_), where the effects are attributed to the second element, and not to strontium [60]. There is limited information regarding the toxicity (LOAEL, HED) of strontium via inhalation; however, according to ATSDR [60], the deposition of strontium particulates in the respiratory tract is dependent on the size of the inhaled particles, and in addition to age, airstream speed, and airway anatomy, and, at least in vitro, strontium can impair the expression of pro-inflammatory cytokines in monocytes [61]. However, it must be considered that we are exposed daily to atmospheric strontium. In Denmark, an adult with a body weight of 70 kg will be exposed to 175 ng of strontium from the air every day [62]. Until there is more information regarding the potential effects of strontium, it will simply stand as an unexpected finding in one of the COVID test swabs.

Finally, titanium and zirconium were detected in three of the COVID test swab brands analysed, and gallium and sulphur were detected in trace amounts, but only in one of the COVID swab brands. Titanium dioxide (TiO_2_) and zirconium silicate (ZrSiO_4_) are manufactured worldwide for use in a wide range of medical, pharmaceutical, and industrial applications [63,64]. Both transition metals have generally been considered toxicologically inert. However, experimental evidence from animal inhalation studies of TiO_2_ nanoparticles has shown that effects are markedly dependent on the model species [65,66], and that in rats, TiO_2_ particles elicit damage via the induction of oxidative stress [63,67,68], even after exposure to repeated daily doses as low as 2 ppm [69] or, even, 1 ppm [70]. Healthy rats exposed to concentrations of ≥2 ppm of TiO_2_ particles develop alveolar macrophage sequestration, focal epithelial hypertrophic and hyperblastic proliferative changes with neutrophilic infiltration, and damage is reversible after exposure stops, as long as the dose was <250 ppm [66]. Zirconium appears to have significantly lower toxicity than titanium [66,71,72,73]. Most of our knowledge on inhalation toxicity derives from animal model studies, and similar studies in humans are still scarce, but there have been reports of non-lethal acute intoxication following inhalation of large amounts of TiO_2_ [74,75], and in vitro studies with human lung cell cultures have shown that there is a marked change in gene expression of cells exposed to TiO_2_, with more than 2000 genes overexpressed, including those related to ROS production [76]. Thus, TiO_2_ nanoparticles are classified as possibly carcinogenic to humans by the International Agency for Research on Cancer [77]. The National Institute of Occupational Safety and Health (NIOSH) recommends to limit aerosol exposure to TiO_2_ particles to <2.4 ppm for fine particles (1–10 μm diameter) and 0.3 ppm for ultrafine particles (<100 nm diameter) as a time-weighted average over a 10 h day and a 40 h work week (OSHA Fact Sheet, retrieved 17 December 2022). Given the small amount of titanium and zirconium detected in the COVID test swabs, its presence would not be expected to cause noticeable damaging effects on the respiratory epithelium of the individuals who undergo swabbing. However, some have suggested that TiO_2_ nanoparticles could exert more damage than previously thought when the compound interacts with metals and other compounds [78], causing oxidative damage to cultured cells even at doses as low as 0.001 µg/mL [79]. These results warrant caution before dismissing the presence of titanium in COVID test swabs.

## 5. Conclusions

In conclusion, we identified eight unexpected chemical elements in the seven randomly selected brands of swabs used for COVID diagnostic tests. The presence and amount of the elements were variable, often being detected in one but not all the replicates of each swab brand, implying that their presence is likely due to quality control errors during manufacturing or packaging, given the unprecedented volume of swabs being produced to meet the demands for COVID testing. We acknowledge the limitations of our study, namely the opportunistic nature of the brands analysed, and the small number of replicates analysed for each brand, although the methods used for analysis are robust, and the presence of the unexpected chemical elements and their amounts detected in the swab heads should not be ignored. If we had detected traces of unexpected chemicals in one or two of the swabs and not in the others, it would have been easier to disregard the finding, but all the swab brands analysed and all the replicates had unexpected chemical elements in their fabric. Furthermore, for some of the elements detected, such as zirconium and titanium, one of the brands exhibited very wide variation in their concentration in the three replicates analysed, suggesting that quality control during swab production may be suboptimal. We believe that having found these chemical elements warrants a careful and systematic evaluation of their presence and amount in a larger number of swabs, which should ideally be conducted by the health authorities, given the widespread use of swabs for nasopharyngeal testing. These elements can induce transient inflammation, lead to cellular stress, deregulate the expression of cytokines, and damage the epithelium following nasal exposure at doses that, at least for some of the elements detected, would be exceeded by exposure to repeated swabbing. Considering the lack of data on the consequences of repeated swabbing of the nasopharyngeal epithelium, and the complete absence of knowledge on the fate of micro- and nanoparticles of the elements identified herein when placed directly on the upper respiratory epithelium, their detection highlights the need for urgent studies. Under the precautionary principle, our findings warrant avoiding the recommendation of repeated testing, particularly of individuals who have no symptoms of COVID, given that, contrary to what was believed at the start of the pandemic, they play a minor epidemiological role [80,81,82,83], particularly at this stage of the pandemic, characterized by markedly lower morbidity and mortality [84]. Having detected these unexpected and potentially toxic chemical elements leads us to propose that, rather than aiding in public health measures, unnecessary frequent swabbing of healthy individuals could jeopardise their health. We are aware that our conclusion can be considered provocative, but we urge public health officials to consider our findings as a justification for urgent studies on the safety of repeated nasopharyngeal swabbing to be conducted, and to recommend against mandatory testing, often required for travelling, attending universities, or as mandatory work policies.

## Figures and Tables

**Figure 1 toxics-11-00805-f001:**
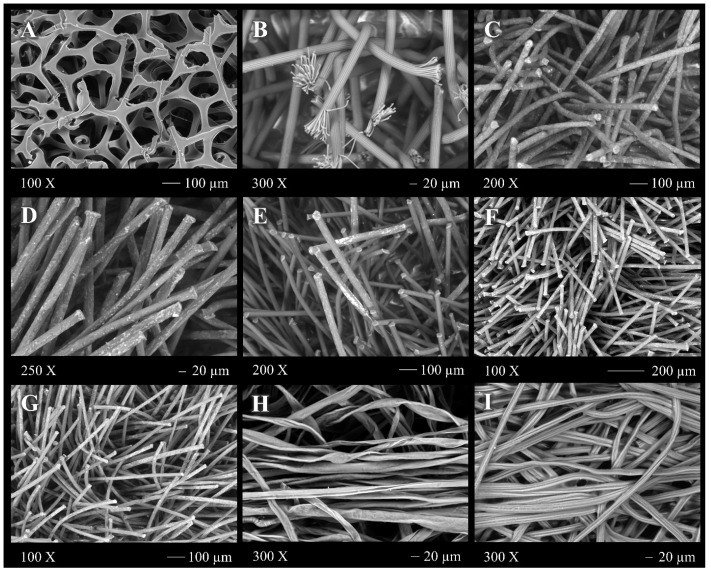
Scanning electron micrographs of different materials and structures of (**A**) foam swab (iHealth), (**B**) flocked nylon swab (Hydraflock), (**C**) flocked nylon swab (nasal swab), (**D**) nylon swab (MANTACC), (**E**) flocked nylon swab (FLOQSwab), (**F**) flocked nylon swab (Kangdaan), (**G**) flocked nylon swab (Taizhou), (**H**) cotton swab (generic applicator, DM Productora), (**I**) cotton swab (COPAN). Magnification was 300× except for panel A, which is 10×.

**Table 1 toxics-11-00805-t001:** Swabs that were examined in this study.

Brand	Fabric	Purpose	Sterilization	CE	Manufacturer	Lot Number
iHealth	Foam	NP	EO	No	iHealth Labs, Inc.	20211213
HydraFlock	Flocked Nylon	NP	EO	Yes	Puritan Med Products, CHN	(10) 50173
Nasal Swab	Flocked Nylon	NP	EO	Yes	CM LAB SAS, BOG and COL	20201221
MANTACC	Nylon	NP	EO	Yes, 0197	Miraclean Technology Co., Ltd., CHN	2021120864
FLOQSwabs	Flocked Nylon	NP	EO	Yes, 0123	COPAN, IT and USA	2010482
Kangdaan	Flocked Nylon	NP	R	Yes, 0197	Shenzhen Kangdaan Biological Technology Co., Ltd., CHN	21CY12001
Taizhou	Flocked Nylon	NP	EO	Yes, 0123	Rich Medical Products, CHN	20220104
Generic	Cotton	A	EO	No	DM Productora S.A. de C.V. MX	070319
Transystem	Cotton	T	R	Yes, 0123	COPAN, IT and USA	211701500

**Table 2 toxics-11-00805-t002:** X-ray (EDX) spectroscopy analysis of the swabs. The table shows the mean and standard deviation of the chemical elements (mg/g) detected for the replicated of each brand. Only one specimen of each of the z swabs was analysed by EDX spectroscopy.

	iHealth	HydraFlock	Nasal Swab	MANTACC	FLOQSwabs	Kangdaan	Taizhou	Generic Applicator	Transystem
Carbon		623.30 ±5.60	576.84 ±59.11	569.97 ±45.99	677.35 ±52.51	611.07 ±4.81	548.70 ±30.26	490.82 NA	509.91 NA
Oxygen	320.69 ±22.69	352.14 ±41.63	363.36 ±2.86	327.33 ±30.43	245.49 ±17.52	245.08 ±2.62	324.98 ±41.61	504.19 NA	470.96 NA
Nitrogen	64.10 ±23.16	- -	46.19 ±3.45	58.45 ±16.87	51.46 ±72.77	101.46 ±2.01	56.59 ±13.98	- -	- -
Fluorine	- -	3.49 ±3.02	2.67 ±3.78	2.28 ±3.95	- -	2.81 ±0.97	5.33 ±0.25	4.99 NA	5.64 NA
Silicon	- -	- -	48.49 ±9.05	43.53 ±7.42	5.78 ±0.54	- -	49.66 ±2.20	- -	- -
Titanium	- -	- -	- -	29.02 ±25.67	5.74 ±0.40	5.93 ±0.86	- -	- -	13.48 NA
Zirconium	- -	- -	- -	2.93 ±5.07	14.20 ±2.91	25.26 ±2.26	- -	- -	- -
Strontium	- -	- -	- -	4.63 ±8.01	- -	- -	- -	- -	- -
Aluminium	- -	- -	7.12 ±1.24	- -	- -	0.03 ±0.04	0.11 ±0.01	- -	- -
Gallium	- -	- -	- -	- -	- -	0.05 ±0.05	- -	- -	- -
Sulphur	- -	- -	- -	- -	- -	0.06 ±0.00	- -	- -	- -
Head Mass	0.0471 g	0.0539 g	0.0468 g	0.0385 g	0.0371 g	0.0493 g	0.0513 g	0.0501 g	0.0575 g

## Data Availability

Data on EDX results and element profiles are available in the Appendix A. Any other data relevant to the manuscript can be requested from the corresponding authors.

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
