# Peer review of "Scanning Electron Microscopy and EDX Spectroscopy of Commercial Swabs Used for COVID-19 Lateral Flow Testing"

_toxics, 2023, doi:10.3390/toxics11100805_

Round 1
Reviewer 1 Report
I find this manuscript interesting and groundbreaking. The ideas presented in this study are very unique and novel. Although the main highlight of this study was to determine the presence of impurities in the swabs, I feel that the authors should have used more than one technique and a sample size of at least 10 in order to authenticate their findings. EDX technique is a good technique but it has its own set of challenges in providing reliable qualitative and quantitative results.
Comments to the authors
This study addresses am in important topic which according to literature has not been fully explored. The authors determined the presents of metals in the Covid test swabs using EDX technique and use the obtained results to predict the effects of the identified metals. I find the motive of this study interesting and worthy to investigate since there is scarce information regarding this topic. The motive of this study is well articulated and supported by facts. I find the study beneficial and it contributes to the dangers associated with metals impurities in the covid swabs. In light of the above, I find the following as areas of improvement for this manuscript. The conclusion is supported with facts drawn from the results and sufficient relevant references have been provided.
1. The authors should have used more than one measurement technique to confirm their results. Quantitative analysis using EDX technique is not reliable.
2. The authors should have used a reasonable size of at least 10 swabs of the same type to validate their claims. One or two samples are not sufficient to justify their claims as the possibilities of bias or external contamination is very high.
3. The authors should have also investigated cases or incidents in which the listed diseases due to these identified elements occurred (maybe a case study or something as evidence).

Author Response
- The authors should have used more than one measurement technique to confirm their results. Quantitative analysis using EDX technique is not reliable.
We agree with the reviewer that more analytical techniques would be ideal. However, our study was conducted as an initial assessment of the elements present in the COVID test swabs. The method used (EM-EDX) is considered suitable to detect the chemical elements, and gives a first approximation of the amount present. Once again, our goal was to identify whether there were unexpected elements in the swabs, and we do not intend for our pilot study to be considered as anything other than that initial approach. We have tried to make this point clearer in the text, and we hope it will inspire more studies in the near future that can use other analytical techniques to more reliably determine the exact amounts of the elements present. However, we urge the reviewer to see our study for what it is, nothing more and nothing less: an initial assessment that could be an important step in understanding what is present in diagnostic swabs, specially when these are mass-produced to meet public health needs. Our study, and the subsequent independent studies that may follow, might help inform manufacturers and lead to the reduction in impurities during their production.
- The authors should have used a reasonable size of at least 10 swabs of the same type to validate their claims. One or two samples are not sufficient to justify their claims as the possibilities of bias or external contamination is very high.
We value the reproducibility of the experiments, and it is why we ran duplicates to attempt to ensure the reliability of the results. However, we cannot include at least 10 replicates for each swab for our pilot study as suggested by the reviewer owing to financial constraints. Once again, we urge the reviewer to see our pilot study for what it is; one that will hopefully help us procuring funds to extend our sample size. Our study is adding to the extremely limited information available about swab composition, and actually, the only other published paper (mentioned in our study) that compared the structure and properties of test swabs used one specimen for each of three brands (https://link.springer.com/content/pdf/10.1007/s10527-021-10120-z.pdf). Far from seeing the paper by Kashapov & Tsibin (2020) as useless, we found the study valuable as it arose our interest in conducting our own study.
- The authors should have also investigated cases or incidents in which the listed diseases due to these identified elements occurred (maybe a case study or something as evidence).
Dada la novedad del descubrimiento, prácticamente no hay datos publicados sobre los efectos de la exposición nasofaríngea a los elementos químicos identificados en seres humanos, y ciertamente no existe ningún informe de caso que sea comparable. Sin embargo, los estudios que comentamos explican la base biológica del daño. Una vez más, nuestro estudio buscó identificar la composición de los hisopos producidos en masa, y esperamos que inspire más trabajo para poder comenzar a registrar incidentes y eventos adversos asociados con estos químicos debido a los hisopos de prueba.
Reviewer 2 Report
In this manuscript, Aparicio-Alonso et al. analyzed the composition of commercial swabs used for the testing of SARS-CoV-2 infections. Although the study is very limited both methodologically and in terms of sample size, it is nevertheless very interesting for the reader as it shows very unexpected and alarming results. It is also interesting because, surprisingly, there are no similar studies to date. The experiments performed are convincing, the tables are clearly arranged. The authors sufficiently discuss and describe the limitations of their study. It is clear that further in-depth and more elaborate studies are needed to shed further and more detailed light on this topic.
Nevertheless, this study should be published, because it could give a reason for these further studies and because it hopefully also leads to the fact that the quality standards at the companies, which manufacture these tests, are raised and it will come in the future to less impurities.
Before a potential publication, the following minor points should still be considered by the authors:
1. Even though the discussion is very informative, it should be shortened by the authors. In relation to the rest of the manuscript, the discussion is too detailed and too long. The most important information about each component found in terms of biological effects should be condensed and tightened more.
2. The sentence on page 2 line 65 "It is sensible to know the components of COVID swabs if we wish to be confident about their safety" should be rephrased, e.g. "to get confident information about the safety of the COVID swabs it is crucial to know their components."
3. Page 2, line 63: „have examined their performance [7,9–11]. and we found only one review paper“, please remove the „.“
4. Page7, line 255 „Regardless, it must be considered that the we are exposed daily to“, please remove „the“.
Some minor points are included in the suggesions for the authors.
Author Response
Thank you very much for taking the time to review this manuscript. Please find the detailed responses below:
- Even though the discussion is very informative, it should be shortened by the authors. In relation to the rest of the manuscript, the discussion is too detailed and too long. The most important information about each component found in terms of biological effects should be condensed and tightened more.
We appreciate the recommendation, and we have reduced the discussion by nearly 300 words and hope the reviewer will find the information more condensed.
- The sentence on page 2 line 65 "It is sensible to know the components of COVID swabs if we wish to be confident about their safety" should be rephrased, e.g. "to get confident information about the safety of the COVID swabs it is crucial to know their components."
The requested change has been done.
- Page 2, line 63: „have examined their performance [7,9–11]. and we found only one review paper“, please remove the „.“
The requested change has been done.
- Page7, line 255 „Regardless, it must be considered that the we are exposed daily to“, please remove „the“.
The requested change has been done.
Round 2
Reviewer 2 Report
The authors have dealt with all concerns I raised. From my point of view, the manuscript can now be published.